# Aetiology and Pathogenesis of Cutaneous Melanoma: Current Concepts and Advances

**DOI:** 10.3390/ijms22126395

**Published:** 2021-06-15

**Authors:** Strahil Strashilov, Angel Yordanov

**Affiliations:** 1Department of Plastic Restorative, Reconstructive and Aesthetic Surgery, University Hospital “Dr. Georgi Stranski”, Medical University Pleven, 5800 Pleven, Bulgaria; 2Clinic of Gynecologic Oncology, University Hospital “Dr. Georgi Stranski”, Medical University Pleven, 5800 Pleven, Bulgaria; angel.jordanov@gmail.com

**Keywords:** aetiology, pathogenesis, melanoma, skin melanoma

## Abstract

Melanoma develops from malignant transformations of the pigment-producing melanocytes. If located in the basal layer of the skin epidermis, melanoma is referred to as cutaneous, which is more frequent. However, as melanocytes are be found in the eyes, ears, gastrointestinal tract, genitalia, urinary system, and meninges, cases of mucosal melanoma or other types (e.g., ocular) may occur. The incidence and morbidity of cutaneous melanoma (cM) are constantly increasing worldwide. Australia and New Zealand are world leaders in this regard with a morbidity rate of 54/100,000 and a mortality rate of 5.6/100,000 for 2015. The aim of this review is to consolidate and present the data related to the aetiology and pathogenesis of cutaneous melanoma, thus rendering them easier to understand. In this article we will discuss these problems and the possible impacts on treatment for this disease.

## 1. Introduction

The term melanoma was first used in 1812 by René Laennec to describe a case of metastatic dissemination of the disease [1]. Cutaneous melanoma (cM) develops from malignant transformations of pigment-producing melanocytes in the basal layer of the skin epidermis. Non-cutaneous melanoma arises from malignantly transformed melanocytes in the uvea [2,3], gastrointestinal tract [4,5], genitalia [6,7], urinary system [8,9], meninges [10,11], etc. The incidence and morbidity of cM are constantly increasing worldwide. Australia and New Zealand are world leaders in this regard with a morbidity rate of 54/100,000 and a mortality rate of 5.6/100,000 for 2015 [12].

The aim of this review is to consolidate and present the data related to aetiology and pathogenesis of cutaneous melanoma, thus rendering them easier to understand. In this article we will discuss these problems and the possible impacts on treatment for this disease.

## 2. Aetiology

The aetiological factors for the development of the cutaneous melanoma are related to both the human body and the environment (Table 1).

### 2.1. Ultraviolet Radiation

Exposure to ultraviolet radiation is the leading risk factor for the development cM. Both natural sunlight and artificial lighting systems are ultraviolet radiation sources [13]. The light spectrum of this radiation includes electromagnetic waves with a length between 200 and 400 nm. The radiation of the UVB wavelength between 290 and 320 nm is the most carcinogenic for the skin [14,15]. Its waves are absorbed at the greatest extent by the nuclear proteins and acids of the skin cells, including melanocytes. They interact directly through oxidative stress, which disrupts the melanocytes. [16,17]. There is also an indirect damaging effect of these rays by the mechanisms of DNA repair. This irreparable damage promotes the occurrence of numerous mutations, inducing carcinogenesis with the transformation of normal cells into cancerous cells [18,19]. 

### 2.2. Skin Phototype

Different skin phototypes are being described and they are dividing the skin of people into six subtypes—I to IV. People with skin phototypes I and II have fair skin, blond or red hair, many freckles, and blue eyes and are most sensitive to ultraviolet exposure. Therefore, these individuals are subject to a higher risk of developing melanoma of the skin due to lower resistance to UVB rays [20,21].

### 2.3. Pigmented Nevi

Pigmented nevi represent benign proliferations of the melanocytes in the skin (Figure 1). 

Pigmented nevi usually remains unchanged in size and behaviour or disappears during life [22]. Approximately 33% of melanomas are derived directly from pigmented nevi [23]. People with multiple different in colour or different in shape pigmented nevi are more prone to developing cM. They are, additionally, found to possess skin phototypes I and II more frequently [24]. 

### 2.4. Use of Pesticides

People working with herbicides are also more likely to develop acral melanoma on the palms of the hands and the soles of the feet. The most commonly used herbicides that result in this disease are the following: dichlorprop, atrazine, propanil, paraquat, diquat, thiocarbamates, alachlor, acetochlor, metolachlor, imazethapyr, pendimethalin, and glyphosate. The incidence of people with acral melanoma who have used these herbicides at home is higher than in people who do not use them [25].

### 2.5. Prolonged Sun Exposure and Sunburn

The World Health Organization declared that prolonged sun exposure and sunburn are carcinogenic to humans. Excessive sun and ultraviolet exposure and frequent sunburns, especially at a younger age, significantly increases the risk of developing cM. [26,27].

### 2.6. Geographical Location

The incidence of cMM varies depending on the geographic location. The highest incidence in the world occurs in Australia and New Zealand [12]. The highest incidence in Europe occurs in Northern Europe and especially Scandinavia. The lowest incidence of cM in Europe is in Eastern and Southern Europe [28].

### 2.7. Genetic Factors

cM develops as a result of the grouping of multiple abnormalities in the melanocyte genetic pathways. They initiate cell proliferation and cessation of apoptosis as a normal response to DNA damage [29,30]. Mutations in various genes are the main driving force in this process. The first gene that is specifically altered in 15–20% of cases is *NRAS* [31,32]. The mutation is found mainly in melanomas, which is caused by prolonged exposure to sunlight, in the nodular form and in melanoma with a greater thickness (>1 mm) [33].

Mutation in the *BRAF* gene is another mutation frequently occurring in cM. It accounts for about 50% of the cases. BRAF kinase plays a role in the regulation of the signalling pathway between mitogen-activated protein kinase and extracellular signal-regulated kinase (MAP/ERK), which controls the cell division and differentiation. As a result, uncontrolled division of melanocytes occurs and results in a development of melanoma [34,35]. This mutation is often found in dysplastic nevi that, once again, demonstrates its role in the carcinogenesis of the disease. It is also found in melanoma cells that are submitted to a prolonged exposure to UV radiation [34]. 

The *PTEN* gene, which encodes a tumour suppressor protein, is a third gene for which mutations may be associated with the development of melanoma. When damaged, the tumour suppressor protein cannot accomplish its role and continuous cell proliferation begins. This mutation is found in 10–20% of patients with primary melanoma [31,36,37].

### 2.8. Heredity

There are several genes (*CDKN2A, CDK4, POT1, ACD, TERF2IP, TERT, BRCA 1 and 2, BAP 1, MITF, TP53, XPC, XPD, XPA,* and *PTEN*) for which mutations may be inherited and results in hereditary melanoma [38]. *CDKN2A*, located on chromosome 9 is gene for which damage is associated with a disorder in the control of cell division. *CDKN2A* mutations are most common in patients with heredity melanoma. It is found in 20% of the families with skin melanoma history. The *CDK4 gene* is located on chromosome 12 and is also involved in regulating cell division. Mutations of this gene are tracked only in isolated cases of cM. *CDKN2A* mutations with p14 loss result in cutaneous melanoma but are also found in pancreatic carcinoma. This explains why some families have an increased risk of developing these two neoplasms [39,40,41,42,43,44,45,46,47]. Mutations in the somatic *gene TP53* encoding protein 53 (p53) also results in the development of heredity melanoma. In its normal state, *p53* functions as a tumour suppressor, permitting damaged cells to recover and halts progression to cancer. In the case of gene mutations, this protein is unable to execute its function [39]. 

### 2.9. Immunosuppressive Conditions

Systemic immunosuppression with cyclosporine and sirolimus in renal transplantation results in the inactivation of tumour suppressors such as p53 and PTCH and the activation of *the proto-oncogenes H-ras, K-ras, and N-ras*. All this results in DNA damage and retention genetic mutation and therefore induces the development and spread of skin melanoma [48]. Patients with immunosuppressive diseases such as AIDS are more likely to develop skin melanoma as the continuous immunosuppression ineffectively protects the organism from occurrence and the progression of many solid tumours [49,50].

### 2.10. Non-Melanoma Skin Cancer

Patients with squamous cell carcinoma, basal cell carcinoma, and actinic keratosis have an increased risk of developing melanoma of the skin. This is most likely due to various genetic factors and prolonged exposure to sunlight [51,52].

## 3. Pathogenesis of Cutaneous Melanoma

The production of melanin by melanocytes is their unique feature. In turn, melanin has a complex of anti-oxidant and pro-oxidant properties [16,53]. Its conversion from an antioxidant to a pro-oxidant agent under the influence of various etiological factors such as UV radiation, heavy metals, herbicides, etc., is the critical and earliest pathogenetic event that initiates carcinogenesis. The pro-oxidant action of the melanin results in an increase in the levels of intracellular oxygen radicals, which in turn causes damage to the DNA molecule of the melanocyte. The result of these mutations is excessive activation of various cell signalling pathways and results in the uncontrolled proliferation, dedifferentiation, and immortalization of specific cell types [54]. 

In 2018, the WHO presented the latest version of the generalized classification of the pathways for the genesis of cutaneous, mucosal, and uveal melanomas based on the main aethiological factor (cumulative sun damage), clinical and pathological characteristics of the precursor lesions, and the genetic aberations which accompany their development. Nodular cutaneous melanoma occupies a special place in this classification because all or most of these pathways refer to it. The pathways which refer to the remaining skin and mucosal melanomas are eight and they are divided into two major groups. The first group includes the melanomas associated with cumulative solar damage (CSD) and consists of three pathways—I, II, III (Table 2).

The second group includes the melanomas that are not associated with cumulative solar damage (no CSD) and consists of five pathways—IV, V, VI, VII, VIII (Table 3) [55].

This classification clearly and accurately describes the pathogenesis of this disease and includes all known genetic mutations involved in it. The exact location of these abnormalities in the various pathogenetic, signaling, and cellular pathways is presented in detail below.

### 3.1. MAPK Pathway

The mitogen-activated protein kinase (MAPK) cascade regulates cell proliferation, growth, and migration and is activated in almost all types of melanomas. This pathway is active under normal conditions, but in the cases of melanoma it is associated with excessive activation [56]. It is activated after binding of growth factors to tyrosine kinase receptors. Stimulation of these receptors activates Ras family proteins—monomeric G proteins (NRAS)—causing the cascade activation of serine/threonine kinases (BRAF) and results in the activation of ERK kinase (also known as MAPK). Serine/threonine kinase activates transcription factors, thereby intensifying the transcription of genes involved in cell growth, proliferation, and migration (Figure 2). 

In 80% of the cases of melanoma or melanocytic nevi, mutations in *NRAS* or *BRAF*
*genes* are observed, which confirms the crucial role of the MAPK pathway [56]. Activating mutations of BRAF serine/threonine kinase (40–50%) and G protein NRAS (15–20%) are the main aberrations leading to the MAPK pathway activation in melanomas [57]. 

Understanding of the MAPK pathway has led to the development of targeted therapy with BRAF inhibitors alone or in combination with MEK inhibitors (vemurafenib + cobimetinib; dabrafenib + trametinib) for patients with activating mutations in *BRAF* [35,58,59].

### 3.2. BRAF

BRAF is a serine/threonine kinase that is activated directly by RAS and is strongly expressed in melanocytes, neural tissue, testes, and hematopoietic cells. BRAF has been found in phosphorylate and activates MEK (a kinase component of the MAPK pathway), which in turn activates ERK (MAPK) by phosphorylation and thus stimulates growth and transformation. This is crucial for the pathogenesis of melanomas [35,59,60]. The conversion of thymidine to adenine (T → A) is the most common mutation in the *BRAF gene* (~70%). It results in the substitution of valine with glutamate *(V600E)* in the protein molecule, resulting in the activation of its kinase domain. These mutations indirectly results in BRAF activation by disrupting the normal intramolecular interactions that hold BRAF in inactive configurations [35,57]. Mutations in the *BRAF gene (V600E)* are more common in melanoma that develops in parts of the body that are exposed to solar radiation. Although *BRAF* mutations may be associated with sun exposure, the transversion (T → A), as mentioned earlier, is not classically related to UV exposure [35,60]. Other amino acid substitutions in the protein are possible, such as those observed in *V600K* mutations (representing ~20% of *BRAF* mutations in melanoma). They are mainly found in melanoma patients exposed to chronic sun exposure. *BRAF V600* mutations represent an early event in the development of melanoma. They are less common in its initial stages—in 10% of the cases with radial growth—and in 6% of in situ melanomas. However, they are very common in metastatic melanoma. These mutations occur in approximately 80% of the benign and the dysplastic nevi, but they alone are not sufficient for the development of cutaneous melanoma [57]. 

Interestingly, *BRAF (V600E)* mutation in nevi involves initial cell proliferation and is followed by oncogene-induced ageing, most likely as a part of the accumulation of *p16 INK4a*; in melanoma *BRAF (V600E)*, mutations only induce uncontrolled proliferation. Many nevi and primary melanomas are polyclonal and contain cells with mutated *BRAF* and wild-type cells (without *BRAF* mutation). Metastatic melanomas, on the other hand, are monoclonal [60]. Mutations in *BRAF V600E* are more common in women and correlate inversely with age [38], while the rate of *BRAF V600K* mutations increases with age [57]. 

### 3.3. RAS

Mutations, leading to increased RAS activity in melanomas also increase cell proliferation, but this occurs significantly less frequently than in other solid tumours [60]. The *Q16R* mutation is the most common *NRAS* mutation in melanoma. Somatic *NRAS gene* mutations may cause increased activity of the NRAS protein, which cannot “shut down”. This results in a serial activation of serine/threonine kinases, stimulating cell cycle progression, cell transformation, and cell survival. The cascade of events can also be caused by overexpression and/or hyperactivation of various growth factor receptors such as c-Met, epidermal growth factor receptor (EGFR), and c-KIT as well as the functional loss of neurofibromatosis type 1 (NF1) tumour suppressor gene, which suppresses NRAS signalling [61,62]. Activating *RAS* mutations were observed in only 10–20% of melanomas (mostly in amelanotic nodular subtypes), with *NRAS* mutations being the most common. *BRAF* mutations only activate the MAPK signalling pathway, while activating *NRAS* mutations simultaneously activates the MAPK and PI3K pathways [60,62]. *NRAS* and *BRAF* mutations have been found to rarely co-occur, indicating that a mutation in one of the two genes is sufficient to activate the MAPK pathway. Activating *BRAF* mutations are more common in nevus cells (70–80% of dysplastic nevi). *NRAS* mutations are rare in nevi and are most common in congenital ones. *NRAS* mutations are often associated with Spitz nevi [60].

The uncontrolled activation of the MAPK signalling pathway in melanomas may be caused by overexpression or hyperactivation of growth factor receptors such as c-Met, c-KIT, and epidermal growth factor receptor (EGFR) [62]. 

### 3.4. c-KIT

c-KIT (tyrosine kinase receptor) and its ligand (stem cell factor) play an essential role in melanocyte development [59,63]. *c-KIT* mutations cause insufficient pigmentation. The results of numerous immunohistochemical studies indicate that the transition from a benign condition to primary or metastatic melanoma is associated with a loss of *c-KIT* expression. The activating mutations and amplification of *KIT genes* have been observed in cutaneous melanomas in areas exposed to chronic sun exposure as well as in acral melanomas (of the hands, feet, and nail bed). *KIT* mutations can activate multiple signalling pathways and the PI3K-AKT pathway in particular. The presence of point mutations in the *KIT gene* has also been observed in gastrointestinal stromal tumours (GIST). The functional characteristics of these mutations are clinically significant as KIT inhibitors are effective in melanomas, but with a much lower degree of clinical response (10–30%) compared with GIST (>70%) [57,63].

### 3.5. c-MET and HGF

Overexpression of another tyrosine kinase receptor c-MET and its ligand HGF (hepatocyte growth factor) correlates with melanoma progression. c-MET is known to control a variety of biological functions such as propagation, survival, motility, and invasion. Tumours can grow and metastasize due to dysregulation caused by aberrant c-MET activation. It should be noted that c-MET (tyrosine kinase receptor) may be overactivated in the event of the excessive secretion of its HGF ligand produced by tumour cells or the tumour microenvironment of melanoma. This paracrine effect activates the PI3K-AKT pathway in tumour cells and leads to resistance to MAPK inhibitors [57,64]. 

### 3.6. Other Factors 

Neurofibromatosis type 1 (NF1) tumour suppressor gene and the negative RAS regulator are other driving factors in the process. *NF1* mutations were identified in 5 of 21 tumours without *BRAF* and *NRAS* mutations. In the context of *BRAF (V600E)* mutations, NF1 has been observed to disrupt normal MAPK and PI3K pathways by inhibiting the development and the metastatic spread of melanoma. Inactivating mutations of neurofibromatosis tumour suppressor *gene 2 (NF2)* have been observed in some melanomas. Initial mutations in *NF1* and *NF2* genes are associated with hereditary neurofibromatosis. Other somatic mutations of the melanoma cells in MAPK downstream effectors have been identified, such as *MAP3K5*, *MAP3K9*, *MEK1*, and *MEK2* [60,65].

### 3.7. PI3K/PTEN/AKT Pathway (Phosphatidylinositol 3-kinase Pathway)

PI3K-AKT is another critical signalling pathway in the cell. It is involved in the regulation of cell survival, growth, and apoptosis [56,66] (Figure 3). 

In carcinomas, this pathway can be genetically driven by activating mutations (e.g., *PIK3CA* and *AKT1*) or by functional loss of some of the components of this pathway (e.g., PTEN). PI3 kinases (phosphatidylinositol 3-kinase-lipid kinases) are triggered directly by tyrosine kinase receptor activation or indirectly by RAS. They cause phosphorylation of PIP2 to PIP3 and subsequent phosphorylation of AKT [57,66]. PTEN lipid phosphatase plays an essential role as an agent antagonizing this pathway by converting PIP3 back to PIP2 [67]. Phosphorylation of AKT (serine/threonine protein kinase) results in the phosphorylation of multi-factor proteins that regulate cellular processes (proliferation, survival, motility, angiogenesis, and metabolism). m-TOR is one of these proteins found to be activated in 73% of human melanoma cell lines and very rarely in nevus cells. Increased activation of PI3K signalling is observed in a large number of melanomas, usually triggered by mutations, deletions, and promoter methylation of the coding genes of the PTEN inhibitor [66]. 

### 3.8. MITF Signalling

Microphthalmia-associated transcription factor (MITF) is a key regulator that is required for melanocyte differentiation, which may affect malignancy in some melanomas. The most common *MITF* genetic alteration is amplification, which occurs in 15–20% of melanomas (more common in metastatic melanomas) [68,69]. These changes are thought to occur later in disease progression and are associated with lower 5 year survival [70]. MITF increases the gene expression involved in the cell cycle progression, the cell proliferation, and the cell survival. It is a transcription factor for cellular cyclin-dependent kinase (CDK2), CDK inhibitors p16INK4a, and p21 as well as for the antiapoptotic mitochondrial membrane protein BCL-2 (B-cell lymphoma 2) [71,72]. *MITF* expression is triggered by the activation of the melanocortin 1-receptor (MC1R), which in turn is activated by the binding of melanocortins (ACTH and α-MSH) [71]. This induces adenylate cyclase activation and the production of c-AMP, which activates protein kinase A (PKA). In turn, PKA activates CREB (cAMP protein-binding response element), which acts as a transcription factor and enhances *MITF* expression. MITF signalling is also closely related and regulated by MAPK signalling. It has been observed that the mutated oncogene *BRAF* can regulate MITF expression, thus ensuring protein levels compatible with the proliferation and survival of melanoma cells. Importantly, MITF may also act as an anti-proliferative transcription factor that induces cell cycle arrest. In melanocytes, *MITF* expression occurs after the binding of the melanocyte-stimulating hormone (MSH) to the melanocortin 1 receptor (MC1R). In this process, MITF targets the genes involved in the regulation of differentiation and pigmentation, distribution, and survival [68,71]. 

### 3.9. p53

*p53* is the primary tumour suppressor gene, associated with apoptosis. The p53 protein encoded is a major transcription factor and bears crucial responsibility for various stresses such as DNA damage, genome instability, hypoxia, and oncogenic aberrations [73]. Disorders in the control of the cell cycle, genomic instability, and abnormal proliferation are observed during the transformation of melanocytes into melanoma cells. Mutations or deletions of p53 are observed in more than 50% of carcinomas. In melanomas, they occur in only 1–5% of primary melanomas and in 11–25% of metastatic melanomas [74]. The expression of the p53 protein in melanomas is variable [75,76], while it is often absent in nevi. Melanoma often results in the overexpression of this protein than when compared to other tumours. Despite the high expression of p53, melanoma cells are highly resistant to apoptosis. This can be explained by the impaired functionality of the p53 apoptotic pathway [77].

### 3.10. Hypoxia-Induced Factor (HIF)

An essential factor for tumour development and progression is the environment itself. Human skin is generally hypoxic, which in turn favours melanogenesis [78]. The hypoxic response is primarily mediated by HIF. It activates many genes, associated with angiogenesis, invasion, and metastasis. Hypoxia in the microenvironment of the skin favours melanocyte transformation and tumour growth induced by the PI3K pathway. Inhibition of the *HIF gene* reduces this process. Tumour hypoxia occurs with melanoma growth due to poor vascularization, which increases HIF and stimulates melanogenesis and melanoma progression, thus being of negative prognostic value [79]. Melanomas located in areas of hypoxia possess the worse prognosis despite treatment [80,81].

### 3.11. Notch Signalling Pathway

Notch signalling pathway has a substantial potential for the development and maintenance of tissue homeostasis. It includes a family of transmembrane receptors and their ligands, negative and positive modifiers, and transcription factors. In mammals, four receptors and five ligands are known [82]. Notch can function as a tumour promoter or suppressor depending on the type of cells. Its activation through the overexpression of *Notch receptor genes 1, 2,* and *4* plays a role in melanoma progression and metastasis, which is a function mediated by β-catenin [83,84,85]. Notch 1 signalling pathway is enhanced in melanoma cells. Along with the AKT pathway and the pathway induced by hypoxia, Notch is involved in the transformation of a normal melanocyte into a melanoma cell [86].

### 3.12. Other Factors Important in the Pathogenesis of Melanoma

Telomerase reverse transcriptase *(TERT)* is a gene located on chromosome 5p15.33 and encodes the catalytic subunit of telomerase. Telomerase is an enzyme that adds nucleotides to telomeres. In normal cells its activity is low, which leads to cell aging and death. The promoter’s mutations of the *TERT gene* are observed in 77% of the pecursor lesions of the melanoma and in a relatively early stage of their genesis, as well as in a large part of the cells of the melanoma itself. This leads to high telomerases activity, which prevents cell aging and apoptosis. The melanomas that cells have mutations in the *TERT gene* are accompanied with a poor prognosis [87].

Numerous cellular interactions, mediated by their cell adhesion molecules (cadherins and adherents) also play an important part in the pathogenesis of melanoma. The immune system also plays a significant role in the carcinogenesis of melanoma. Essential factors for anti-melanoma immunity are two types of immune response: humoral and cell-mediated. In order to survive, cancer cells can modify the immune response by employing several mechanisms such as reduction or deactivation of antigen presentation, immunological barriers in the tumour microenvironment, negative regulatory pathways in T cells, and T cell dysfunction [50,88,89]. Cytotoxic T lymphocyte-associated antigen 4 (CTLA-4) and programmed cell death protein 1 (PD1) are the two immune checkpoints that regulate immune homeostasis by inhibiting T cell activation. The development of monoclonal antibodies directed against CTLA-4 (ipilimumab) and PD1 (nivolumab, pembrolizumab), which eliminates the inhibition of T cell activity and restores the recognition ability of T cells, is a game-changer in the treatment of this disease; the development provides better treatment response and improved long-term survival [50]. These developments have become the gold standard for treatment in most patients with advanced and metastatic melanoma and are currently used in many other solid cancers and hematological malignancies [5,90,91]. 

A distinctive feature of the malignant neoplasms is their ability to disseminate and metastasize. Typically, melanocytes bind to basal keratinocytes through cell–cell adhesion molecules, such as the transmembrane glycoprotein E-cadherin. Loss of E-cadherin and upregulation of N-cadherin leads to the separation of melanoma cells from the epidermis and promotes melanoma invasion. Phosphatase and tensin homologue (PTEN) are suggested as potential regulators in the cell adhesion process [67,92]. 

## 4. Conclusions

This summary of the scientific data on the aetiology and pathogenesis of cutaneous melanoma is useful for understanding the basic features of melanoma, which is an important oncological entity. Despite significant recent progress in the management of this disease during the last couple of years, there is still a need for improvements in the prevention and the treatment of this deadly disease. Hopefully, this may result in the further reduction in the morbidity and the mortality of melanoma worldwide.

## Figures and Tables

**Figure 1 ijms-22-06395-f001:**
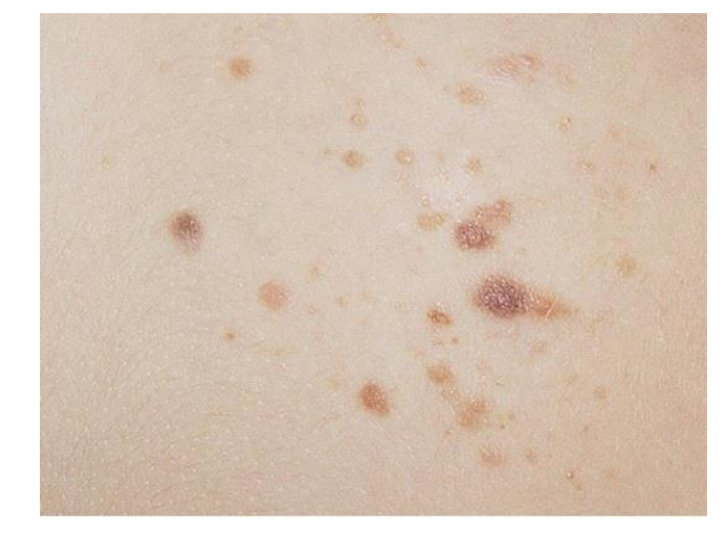
Pigmented nevi.

**Figure 2 ijms-22-06395-f002:**
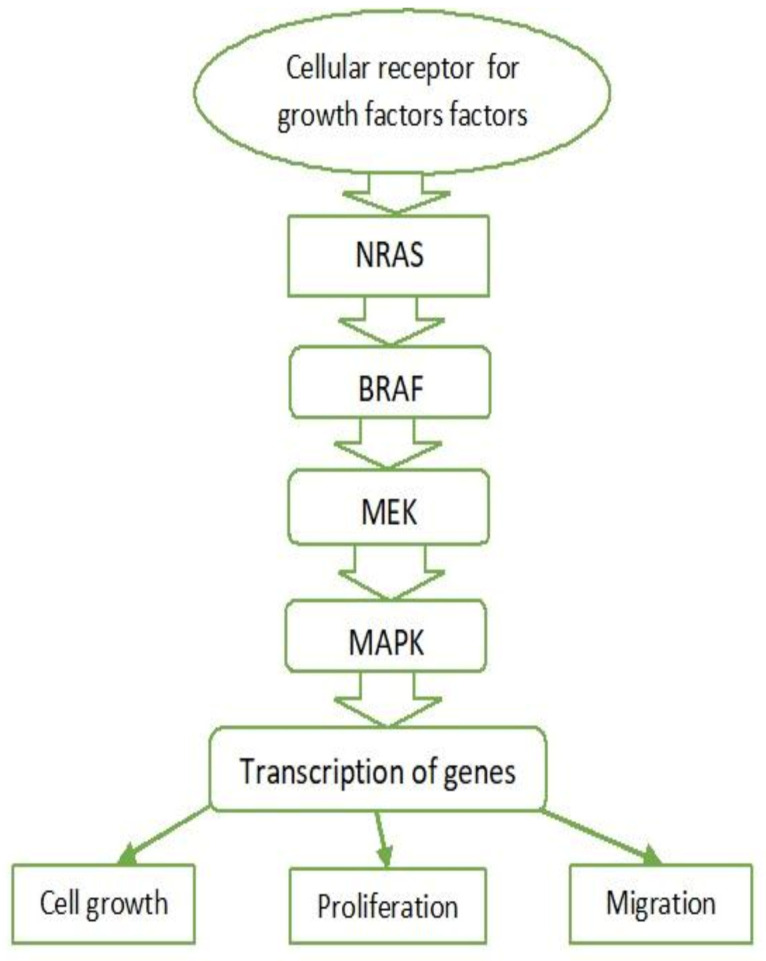
MAPK pathway. NRAS—Neuroblastoma RAS viral oncogene homolog. BRAF—v-Raf murine sarcoma viral oncogene homolog B. MEK—Mitogen-activated protein kinase kinase. MAPK—Mitogen-activated protein kinase.

**Figure 3 ijms-22-06395-f003:**
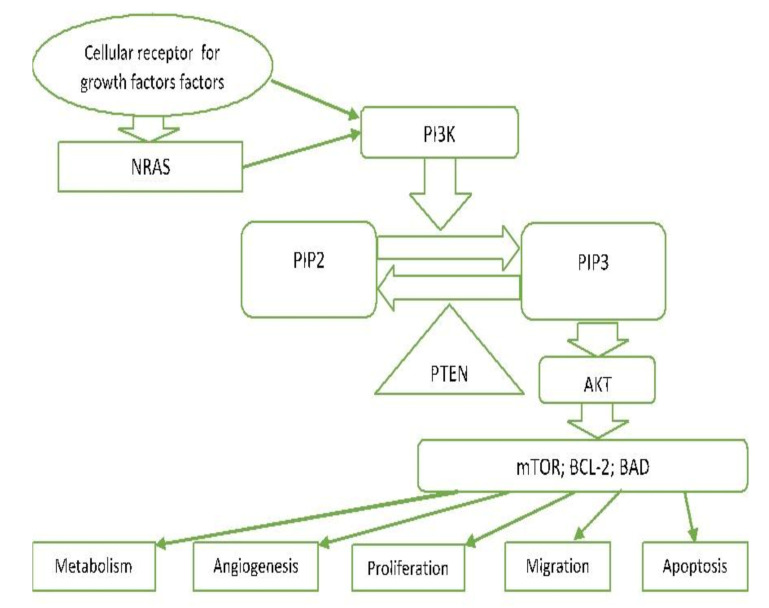
PI3K/PTEN/AKT pathway. NRAS—Neuroblastoma RAS viral oncogene homolog. PI3—Phosphoinositol-3-kinase. PIP2—Phosphatidylinositol 4,5-bisphosphate. PIP3—Phosphatidylinositol (3,4,5)-trisphosphate. PTEN—Phosphatase and tensin homolog deleted on chromosome 10. AKT—Protein kinase B. mTOR—The mechanistic target of rapamycin. BCL-2—B-cell lymphoma-2. BAD—proapoptotic protein.

**Table 1 ijms-22-06395-t001:** Risk factors for the development of cutaneous melanoma (cM).

Risk Factors Related to Human	Risk Factors Related to the Environment
Skin phototype Pigmented nevi Genetic factors Heredity Immunosuppressive conditions Non-melanoma skin cancer	Ultraviolet radiation Use of pesticides Prolonged sun exposure and sunburn Geographical location

**Table 2 ijms-22-06395-t002:** Pathways for melanomas typically associated with cumulative solar damage.

Melanomas Typically Associated with Cumulative Solar Damage			
**Pathway**	**I** SSM/ low-CSD melanoma	**II** LMM/ high-CSD melanoma	**III** Desmoplastic melanoma
**Benign** **Neoplasms (nevi)**	Nevus	IMP (?)	IMP (?)
**Intermediate** **Neoplasms** (low-grade dysplasias and melanocytomas)	^1^ Nevus with low-grade dysplasia; ^2^ BIN; ^3^ DPN	IAMP/dysplasia (?)	IAMP/dysplasia (?)
**Intermediate** **neoplasms** (high-grade dysplasias and melanocytomas)	^1^ Nevus with high-grade dysplasia/ MIS; ^2^ BAP1- inactivated melanocytoma/ MELTUMP; ^3^ Deep penetrating melanocytoma/ MELTUMP; ^4^ PEM/ MELTUMP	Lentigo maligna (MIS)	MIS
**Malignant neoplasms**	^1^ SSM(VGP); ^2^ Melanoma in BIN(rare); ^3^ Melanoma in DPN(rare); ^4^ Melanoma in PEM(rare);	LMM (VGP)	Desmoplastic melanoma
**Common mutations**	^1^ BRAF p.V600 ^b^; NRAS ^b^; TERT ^d^; CDKN2A ^a^; TP53 ^a^; PTEN ^a^^2^ BRAF ^b^; NRAS ^b^ + BAP1 ^a^; ^3^ BRAF ^b^; MAP2K1 ^b^; NRAS ^b^ + CTNNB1 ^b^; APC ^a^;^4^ BRAF ^b^ +PRKAR1A ^a^; RKCA ^c^	NRAS ^b^; BRAF non-p.V600E ^b^; KIT ^b^; NF1 ^a^; TERT ^d^; CDKN2A ^a^; TP53 ^a^; PTEN ^a^; RAC1 ^b^	NF1 ^a^; ERBB2 ^e^; MAP2K1 ^e^; MAP3K1 ^e^; BRAF ^e^; EGFR ^e^; MET ^e^; TERT ^d^; NFKBIE ^d^; NRAS ^b^; PIK3CA ^b^; PTPN11 ^b^

Abbreviations: BIN, BAP1-inactivated nevus; CSD, cumulative solar damage; DPN, deep penetrating nevus; IAMP, intraepidermal atypical melanocytic proliferation; IMP, intraepidermal melanocytic proliferation without atypia; LMM, lentigo maligna melanom; MELTUMP, melanocytic tumor of uncertain malignant potential; MIS, melanoma in situ; PEM, pigmented epithelioid melanocytoma; SSM, superficial spreading melanoma; VGP, vertical growth phase (tumorigenic and/or mitogenic melanoma). ^a^ Loss-of-function mutation; ^b^ gain-of-function mutation; ^c^ rearrangement; ^d^ promoter mutation; ^e^ amplification. ^1,2,3,4^ Kinds of SSM/ low-CSD melanoma. (?) probable pathological type.

**Table 3 ijms-22-06395-t003:** Pathways for melanomas that are not consistently associated with cumulative solar damage.

Melanomas Not Consistently Associated with Cumulative Solar Damage					
**Pathway**	**IV** Spitz melanoma	**V** Acral melanoma	**VI** Mucosal melanoma	**VII** Melanomas arising in congenital nevi	**VIII** Melanomas arising in blue nevi
**Benign** **neoplasms (nevi)**	Spitz nevus	Acral nevus (?)	Melanosis (?)	Congenital nevus	Blue nevus
**Intermediate** **neoplasms** (low-grade dysplasias and melanocytomas)	Atypica Spitz tumor melanocytoma)	IAMPUS/dysplasia	Atypical melanosis /dysplasia/IAMPUS	Nodul in congenital nevus (melanocytoma)	(Atypical) cellular blue nevus (melanocytoma)
**Intermediate** **neoplasms** (high -grade dysplasias and melanocytomas)	STUMP/ MELTUMP	Acral MIS	Mucosal MIS	MIS in congenital nevus	Atypical cellular blue nevus
**Malignant neoplasms**	Malignant Spitz tumor/Spitz melanoma (tumorigenic)	Acral melanoma (VGP)	Mucosal lentiginous melanoma (VGP)	Melanoma in congenital nevus (tumorigenic)	Melanoma in blue nevus (tumorigenic)
**Common mutations**	HRAS ^b^; ALK ^e^; ROS1 ^e^; RET ^e^; NTRK1 ^e^; NTRK3 ^e^; BRAF ^e;^ MET ^e^; CDKN2A ^a^	KIT ^b^; NRAS ^b^; BRAF ^b^; HRAS ^b^; KRAS ^b^; NTRK3 ^e^; ALK ^e^; NF1 ^a^; CDKN2A ^a^; TERT ^f^; CCND1 ^d^; GAB2 ^d^	KIT ^b^; NRAS ^b^; KRAS ^b^; BRAF ^b^; NF1 ^b^; CDKN2A ^a^; SF3B1 ^a^; CCND1 ^d^; CDK4 ^d^; MDM2 ^d^	NRAS ^b^; BRAF p.V600E ^b^ (small lesions); BRAF ^e^	GNAQ ^b^; GNA11 ^b^; CYSLTR2 ^b^; BAP1 ^a^; EIF1AX ^c^; SF3B1^c^

Abbreviations: IAMPUS, intraepidermal atypical melanocytic proliferation of uncertain malignant potential; MELTUMP, melanocytic tumor of uncertain malignant potential; MIS, melanoma in situ; STUMP, spizoid tumor of uncertain malignant potential; VGP, vertical growth phase (tumorigenic and/or mitogenic melanoma). ^a^ Loss-of-function mutation; ^b^ gain-of-function mutation; ^c^ change-of-function mutation; ^d^ amplification; ^e^ rearrangement; ^f^ promoter mutation. (?) probable pathological type.

## Data Availability

Please refer to suggested Data Availability Statements in section “MDPI Research Data Policies” at https://www.mdpi.com/ethics.

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
