# Peer review of "Aetiology and Pathogenesis of Cutaneous Melanoma: Current Concepts and Advances"

_ijms, 2021, doi:10.3390/ijms22126395_

Round 1

Reviewer 1 Report

  The authors reviewed the data related to aetiology and pathogenesis of cutaneous melanoma and discussed the problems of this disease.

There are several points to be answered.

  1. I think experts use just ‘melanoma’ instead of ‘malignant melanoma’ recently.
  2. To discuss about pathogenesis of melanoma, especially from the viewpoint of genetics, WHO classification published in 2018 can not be ignored.
  3. The authors described that people working with herbicides are more likely to develop cMM on the palms of the hands and the soles of the feet. Are they cutaneous melanoma, or acral melanoma? Usually palms and soles are frequent sites of acral melanoma.

Reviewer 2 Report

This is a comprehensive review which will bring readers up to current information about melanoma. Reference could be made to studies on TERT. N Engl J Med 2015; 373:1926-1936

The review is quite good and I recommend it be published

Round 2

Reviewer 1 Report

The authors responded adequately.

Author Response

Thank you !!!

Sincerely

Assoc. Prof. Dr. Strahil Strashilov PhD